# Enhanced Classification of Dog Activities with Quaternion-Based Fusion Approach on High-Dimensional Raw Data from Wearable Sensors

**DOI:** 10.3390/s22239471

**Published:** 2022-12-04

**Authors:** Azamjon Muminov, Mukhriddin Mukhiddinov, Jinsoo Cho

**Affiliations:** Department of Computer Engineering, Gachon University, Seongnam 13120, Republic of Korea

**Keywords:** activity classification, animal behavior, smart costume, sensors, pet care, IoT, machine learning, SVM

## Abstract

The employment of machine learning algorithms to the data provided by wearable movement sensors is one of the most common methods to detect pets’ behaviors and monitor their well-being. However, defining features that lead to highly accurate behavior classification is quite challenging. To address this problem, in this study we aim to classify six main dog activities (standing, walking, running, sitting, lying down, and resting) using high-dimensional sensor raw data. Data were received from the accelerometer and gyroscope sensors that are designed to be attached to the dog’s smart costume. Once data are received, the module computes a quaternion value for each data point that provides handful features for classification. Next, to perform the classification, we used several supervised machine learning algorithms, such as the Gaussian naïve Bayes (GNB), Decision Tree (DT), K-nearest neighbor (KNN), and support vector machine (SVM). In order to evaluate the performance, we finally compared the proposed approach’s F-score accuracies with the accuracy of classic approach performance, where sensors’ data are collected without computing the quaternion value and directly utilized by the model. Overall, 18 dogs equipped with harnesses participated in the experiment. The results of the experiment show a significantly enhanced classification with the proposed approach. Among all the classifiers, the GNB classification model achieved the highest accuracy for dog behavior. The behaviors are classified with F-score accuracies of 0.94, 0.86, 0.94, 0.89, 0.95, and 1, respectively. Moreover, it has been observed that the GNB classifier achieved 93% accuracy on average with the dataset consisting of quaternion values. In contrast, it was only 88% when the model used the dataset from sensors’ data.

## 1. Introduction

Dogs are the most common domestic pets in many parts of the world today [1]. Since many dog owners treat their dogs like family members, the health of the dog is often intrinsically linked to the health of its owner. With canine health in mind, behavioral classification plays an important role in monitoring dogs, making medical diagnoses, and ensuring animal welfare. Dog activity classification methods using sensory technologies, such as gyroscopes, magnetometers, and accelerometers, have enabled researchers to better understand their behavior, locomotor activity, and social interactions [2,3,4,5]. A market report has reported a combination of a smartphone application and sensor data as an IoT-based solution for monitoring pet health [6].

Accelerometry has recently been introduced in veterinary medicine as a new method to objectively assess the activity level of a dog [7,8,9]. The method is an indispensable tool often used in combination with other previously tested performance indicators such as ground reaction forces [10,11,12,13,14]. Omnidirectional activity monitoring measures spontaneous activity during the control period and determines the associated acceleration sources [14]. Activity monitoring has been used to objectively determine the response to drug therapy and to evaluate a dog’s energy expenditure [15,16]. The use of accelerometry for a large objective analysis of dogs has great potential in clinical practice. However, it is impractical to obtain data on the kinetic activity of dogs [12].

On the one hand, the reaction force parameters of dogs, which are fully normalized to body size and body weight according to the theory of dynamic similarity, differ for different dog breeds [17]. On the other hand, with modern technology, we can see that the “Pet Pace” system provides an extensive database for each patient. Monitoring these parameters, analyzing them, and detecting deviations from typical patterns provide real-time indications of susceptibility to various health problems that occur in different dogs.

In this study, we tried to find an answer to whether there is another way to classify the dog’s behaviors with high accuracy. To this end, we propose to obtain the data from the sensors, compute moving measurement values called quaternion values, and perform the classification based on these quaternion values. We also compared machine learning (ML) algorithms based on the proposed approach, such as the K-nearest neighbor, RBM core support vector machine, Decision Tree, and Gaussian naïve Bayes. This method is based on extracting the correct source data for the training models of ML algorithms, calculated using sensors such as accelerometers and gyroscopes. The accuracy of the trained model was demonstrated in the experiments. The authors believe the current approach can improve classification accuracy in real-time deployment scenarios and dog welfare. In this study, we attempted to use a few sensors and simple algorithms to avoid hardware problems.

We summarize our main contributions in more detail as follows:(a)First, we study the quaternion values’ role in the classification task. We create six datasets that describe a dog’s particular activity: standing, walking, running, sitting, lying down, and resting, from the data received by the accelerometer and gyro sensors. Next, we compute the quaternion values for each data point in the six datasets. These values serve to train four different machine learning models that work on four different algorithms: GNB, DT, KNN, and SVM. Finally, by comparing four different classification F-scores, we choose the model that has the highest accuracy.(b)Second, we develop a low-cost smart costume for dogs that requires minimum hardware employment and obtains a highly accurate behavior classification. The smart costume involves the following: Raspberry Pi Zero as a processing unit; a real-time clock to determine an appropriate time for behavior; and a communication module to send the data to the ML model, accelerometer, and gyro sensors to obtain the data, battery, and memory for the storage of datasets. Using this, once we train, test, and choose the best performance ML model, we can upload the trained model to the processing unit, and the smart costume starts classification on its own without communication of any outsources, which overcomes the loss-of-communication problem.(c)Finally, we detect the data features that enhance the classification accuracy. The ML model receives two different datasets that describe the same activity. One dataset consists of only accelerometer and gyro sensor data, whereas the other one consists of quaternion values that are denoted based on the sensor’s data. With two different datasets, the model gives two different classification outputs. By comparing the F-scores that come from both outputs, we come to a conclusion about if our approach improves the accuracy or not.

The remainder of this paper is organized as follows. Section 2 provides the review of the relevant literature, and Section 3 gives detailed information about the materials and methods. It also details the construction of the smart costume hardware and feature extraction. Section 4 describes the experimental methods and results. Finally, Section 5 presents the findings.

## 2. Related Works

In recent years, the number of consumer-oriented activity trackers for dogs has expanded, and the industry is likely to grow fast in the following years. Dog owners and their canine companions can benefit from more precise and comprehensive monitoring of the dog’s movements and body postures. Detailed monitoring of a dog’s daily activities would enhance dog owners’ comprehension of certain canine behaviors and reactions, such as separation anxiety when left alone at home. Moreover, activity recognition plays an important role in the search and rescue (SaR) field. Kasnesis et al. performed activity recognition using deep learning based on sensor data for SaR [18]. The ability to identify more specific behaviors from accelerometer data can be utilized as an indicator of the animal’s well-being and health state, for instance, by recognizing stress- and pain-related behaviors [19,20]. Commercialized devices allow researchers to track dog’s activity, which can lead to improving their health [21]. Automatic behavior differentiation would also improve behavioral research, where behaviors are usually recorded by labor-intensive and time-consuming manual annotation of video recordings. It would enable a more thorough behavioral study on free-roaming wild animals [22]. In research employing these video data, overlap with other objects drastically reduced behavior detection accuracy. In addition, because behavior identification is only feasible inside the camera-visible radius, it is challenging to shoot dogs at home without capturing the entire residence with cameras.

The majority of commercially available dog wearable gadgets have an accelerometer sensor to measure the dog’s behavior [23]. Due to the employment of a single sensor, the activity level of dogs is defined into a few classifications as opposed to more complex behavior identification. Integrating and analyzing gyroscope sensor data has lately made more precise behavior detection feasible, although achieving high recognition accuracy for all activities takes time and effort. Recent research has thus been performed to identify dog behavior based on data from wearable sensors such as accelerometers and gyroscopes. Ladha et al. [24] distinguished 16 canine actions in realistic contexts with a global accuracy of 68.6%. Using a hierarchical “one vs. the rest” classifier, van Uijl et al. [25] demonstrated that walk, trot, canter/gallop, eat, drink, and headshake behaviors could be categorized with 95% accuracy. In reference [26], a three-axis accelerometer and three-axis gyroscope were used to differentiate seven canine actions, including sitting and trotting, with an accuracy of more than 80%. In identifying eight behaviors, Ferdinandy et al. [27] attained up to 60% or 80% accuracy depending on the cross-validation approach, and Ladha and Hoffman [28] achieved 86% accuracy in recognizing the dog resting. By embedding various sensors, such as a gyroscope and inertial measurement unit (IMU) in the wearable device, Chambers et al. [29] were able to validate the sensitivity and specificity for detecting specific behavior such as drinking and eating.

Very similar research to the current study has been done by Kim et al. [30], where they proposed multimodal data-based behavior recognition of dogs by fusing sensor data (accelerator, gyroscope) obtained from a wearable device and video data received from a camera. They aimed to classify seven types of dog behaviors (standing, sitting, lying with and without a raised head, walking, sniffing, and running), and they reported 91% accuracy on average.

The similarity of the above studies is that they obtained their behavior classification results by relying on the data provided by wearable sensors. Although the sensor data play the most crucial role in recognizing a dog’s behavior, we apply additional calculations on the sensor data to extract the valuable features that lead to high-accuracy classification. To the best of our knowledge, our work is the first study on increasing the behavior recognition accuracy by fusing the accelerometer and gyro sensors’ data with quaternion degree values.

## 3. Materials and Methods

This section consists of two sub-sections, where one briefly explains the design of the low-cost smart costume, and another one unfolds the process of computing the quaternion values and behavior classification performance.

### 3.1. Sensor-Based Smart Costume

We develop a low-cost smart costume that is worn on the body of a dog. Figure 1a shows the architecture of the smart costume hardware system. Figure 1b shows a fully assembled smart costume for a dog.

The smart costume consists of a dog’s suit and a central shield, which contain the following:Processing unit: The Raspberry Pi Zero is a small, classic Raspberry breadboard. The Raspberry Pi Zero is equipped with an ATmega328 microcontroller, which runs at 16 MHz and has 32 KB flash memory. It has a small footprint with a length of 45 mm and a width of 18 mm. Being the smallest Raspberry Pi Zero in the Raspberry family, the weight of the Raspberry Pi Zero is 7 g.Real-time clock: The RTC module is used for up to date the time and date. In the current study, we used the date and time to separate the data from the sensors.Communication module: We used HC-05 Bluetooth as the communication module in this study. HC-05 Bluetooth is an easy-to-use Bluetooth serial port protocol module designed to establish a transparent wireless serial connection setup. The current model communication is based on Bluetooth Low Energy, which can be conveniently used with other iOS devices simultaneously and has low power requirements. The HC-05 Bluetooth module provides a switching mode between the master and slave modes, which means that it cannot use either the received or transmitted data.Sensors: The MPU-6050 devices combine a three-axis gyroscope and three-axis accelerometer on the same silicon die, together with an onboard Digital Motion Processor™, that processes complex six-axis MotionFusion algorithms. It also has the added feature of an integrated temperature sensor. The device can access external magnetometers or other sensors through an auxiliary master I²C bus, allowing the device to acquire a full set of sensor data without intervention from the system processor. The devices are offered in a 4 mm × 4 mm x 0.9 mm QFN package. In the current study, the MPU-6050 module played a crucial role in achieving real-time activity.Memory: we used a micro-SD card reader for the memory card on which the sensor data are stored.Battery and charger: To make a complete project, we added a 3.7 v Li-ion battery with a PowerBoost 1000C charger. This small DC/DC boost converter module can be powered by any 3.7 V LiIon/LiPoly battery and converts the battery output to 5.2 V DC for use in our 5 V project.

### 3.2. Physical Activity Classification

**Feature Extraction**. The aim of this study is to find a way of enhancement the accuracy of dog behavior recognition. Therefore, we intend to classify the six categories of dog activities, and Table 1 summarizes the full description of the behavior denotations.

To classify the activities, we use sensor data and an independent source that can verify the activities of the dogs. For the beginning, we obtain six-dimensional raw data (triaxial accelerometer data and triaxial gyro-sensor data) from the dog’s smart costume. The raw data are denoted, and the three coordinates of the acceleration are as follows: $\left[acc_{x}\left(t\right),\;acc_{y}\left(t\right),\;acc_{z}\left(t\right)\right]$ and three angular velocity coordinates that receives from gyro-sensor $[\omega_{x}\left(t\right),\;\omega_{y}\left(t\right),\;\omega_{z}\left(t\right)].$ The use of conventional acceleration and gyroscope raw data was not sufficient to determine the critical poses of a dog, which led to reduction the classification accuracy. In order to enhance the accuracy, we denote the new rotation values called quaternion values [16,31] that provide a real-time estimation of the orientation by utilizing the raw data. 

Starting from a state of rest, the acceleration acc (0) was equal to gravity, providing the vertical axis *z*. The *xyz* reference was defined as the rotation of *xyz*(0) around the horizontal axis V(0) that aligns the vertical axis with the *z* direction. At time 0, the inclination θ(0) of *z* may be represented as:(1)$$\theta \left(0\right)=cos^{-1}\left(-acc\left(0\right)•z\right)=cos^{-1}\left(acc_{z}\left(0\right)\right)$$
(2)$$V\left(0\right)=-acc\left(0\right)\;✕\;z=\left[-acc_{y}\left(0\right),acc_{x}\left(0\right),0\right]$$
where • and ✕ denote the standard vector dot and cross-product, respectively.

The starting orientation of the segment ***q***(**0**) in *xyz* equates to using quaternion notation:(3)$$q\left(0\right)=\left[cos\left(\frac{\theta \left(0\right)}{2}\right),sin\left(\frac{\theta \left(0\right)}{2}\right)✕\;\left[\frac{V\left(0\right)}{\|V\left(0)\right\|}\right]\right]$$

As a result, the orientation of the segment relative to the fixed reference was improved for each sample (*t* = 1, 2, …) by integrating the angular velocity ω(t). By making the assumption that the sample frequency f was sufficiently high to have small rotation and a constant angular velocity between two consecutive samples, the new orientation ***q(t)*** was calculated [17]:(4)$$\{\begin{array}{c} \Omega \left(t\right)=q\left(t-1\right)⊗\left(\frac{\omega \left(t\right)}{f}\right)⊗q{\left(t-1\right)}^{-1}\\ \Phi \left(t\right)=\left[1,\frac{\left[\Omega_{x}\left(t\right),\Omega_{y}\left(t\right),\Omega_{z}\left(t\right)\right]}{2}\right]\;\\ q\left(t\right)=\Phi \left(t\right)⊗q\left(t-1\right)\; \end{array}$$
where ⊗ denotes the product operator associated with quaternions, and Φ(t) corresponds to the quaternion that rotates the orientation from q(t−1) to q(t).

**Behavior classification**. As the computing of quaternion values is finished and stored into one dataset, we employ the k-fold cross-validation technique to split the entire dataset into *k* sub-datasets (in this case *k* = 10) with the *k − 1* part for training and the remaining for testing the performance. Finally, a classification model that classifies the six different dog activity behaviors seeks to classify the behaviors by using the classifier algorithm. Since the number of behaviors is six, we chose the following multi-class classifiers that suit our task and are the most common.

**Support vector machine**. In the presence of a nonlinear relationship between the characteristics and the response, the quality of the linear classifiers can often be unsatisfactory. To account for the nonlinearity, the space of variables is usually expanded, including various functional transformations of the original predictors (e.g., polynomials and exponentials). The SVM can be considered a nonlinear generalization of the linear classifier based on the extension of the dimension of the original predictor space using a particular kernel function. This allows for the creation of models by dividing the surfaces into different shapes. In this study, we used the radial basis function (RBF) as the kernel function. To fit the RBF kernel SVM model, we first need to evaluate the values of two parameters: C (cost) is the allowable penalty for violating the boundary of the gap, and γ (gamma) is the parameter of the radial function [32]. In our experiments, we used the search function on the SVM grid of the scikit-learn library in Python [33]. The function finds the effective gamma and C parameters of the RBF. The best parameters are defined by a function: ${\{}^{‘}C^{’}:10.0,‘gamma’:1.0\}$ with a total score of 0.91.**K-nearest neighbor**. KNN is a metric method used for autonomous object classification and regression. In the classification procedure, the item is given to the class with the highest frequency among its neighbors whose classes are already known. Using the regression method, the item is assigned the average value of the K closest objects whose values are known. Only the K value is needed for the current classification. We used a typical starting value for *K* as $K=\sqrt{N}$ as the optimal value, where N is the number of elements in the training dataset [34].**Decision Tree**. DT represent the rules in a hierarchical sequential structure, with each object corresponding to one node that provides a solution. DTs are excellent for classifying tasks, that is, assigning objects to one of the previously known classes. The target variable must have discrete values [35]. The tree structure consists of “leaves” and “branches”. The edges (“branches”) of the DT contain attributes on which the objective function depends. The values of the objective function are recorded in the “leaves,” and the attributes that distinguish the observations are recorded in other nodes. To classify a new case, we need to go down the tree to the sheet and give the corresponding value. We used a value of 5 as the maximum depth of the tree. Similar decision trees are often used in data mining.**Gaussian naïve Bayes**. The naïve Bayes classifier is a straightforward probabilistic classifier based on Bayes’ theorem with tight (naïve) independence assumptions. Depending on the precise characteristics of the probabilistic model, naïve Bayes classifiers can be efficiently trained. In numerous practical applications, maximum likelihood is employed to estimate the parameters of naïve Bayesian models. In other words, it is possible to utilize a naïve Bayes model without believing in Bayesian probability or employing Bayesian procedures. Despite their unsophisticated appearance and undeniably very simple conditions, naïve Bayes classifiers perform far better in many challenging real-world circumstances. A benefit of the naïve Bayes classifier is that a modest quantity of training, parameter estimate, and classification data can be employed [36]. The GNB algorithm employs a probabilistic method. It entails an initial and subsequent computation of the likelihood of classes within the dataset and the test data for that class. Typically, when working with continuous data, it is believed that the continuous values associated with each class follow a normal (or Gaussian) distribution [36].



(5)
$$P\left(x_{i}|c\right)=\frac{1}{\sqrt{2\;*\;\pi \;*\;sigma_{x_{i,c}}^{2}}}\;*\;\exp\left(-\frac{{\left(x_{i}-mean_{x_{i,c}}\right)}^{2}}{2\;*\;sigma_{x_{i,c}}^{2}}\right)$$



A mathematical expression for determining the conditional probabilities of a test feature given a class is shown in Equation (2). Here, $x_{i}$ is a test data feature, c is a class, and $sigma^{2}$ is the associated sample variance.

The visual illustration of the general process of the sensor’s data acquisition, data processing, and dog activity classification is shown in Figure 2. 

## 4. Experiment and Results

In this section, we present the data collection process for performing two different experimental results to detect the valuable feature data that leads to accurate behavior classification. We conducted the first experiment using the quaternion data and the second experiment using the accelerometer and gyro sensor data directly to make a classification. Thereafter, by obtaining the F-scores of the two experiments, we detected the role of the quaternion data in the behavior classification task. 

### 4.1. Data Collection

The subjects of the current study were 18 healthy adult domestic dogs with medium and large (11 males and 7 females with an average age of 4 years) sizes. The average weight of the dogs is 15.8 kg (12–18 kg). They are widespread Korean domestic dogs, and detailed information about them is listed in Table 2.

We chose domestic dogs among university students with the permission of their owners. Over three days, we observed and collected sensor data received from 18 dogs’ smart costumes. The experiment was conducted on a football playground, and the smart suit was programmed to transmit data at a frequency of 1 Hz (one row of data per second). During the experiment, we recorded the experimental area with a surveillance camera to verify the dogs’ activities. The recording plays an essential role by defining the exact period of the particular behavior, and we included all sensor data into the corresponding behavior dataset. For example, if we detected that dog is in a lying position from 9:40 am to 9:55 am, then we included all the accelerometer and gyro sensors’ data received within this period into the “lying” dataset. The smart costume recorded every per second data point from 8 a.m. to 6 p.m. To avoid inconsistencies among datasets, we paid attention to the amount of all the raw data being the same throughout the datasets. The experimental model structure is shown in Figure 3.

**Performance Evaluation**. To evaluate the effectiveness of the classifications, we used the F-score, feedback (recall), and accuracy (precision) [20] as evaluation metrics.
(6)$$F-score=\frac{2\;*\;P\;*\;R}{P+R}\;$$(7)$$Recall=\frac{truepos}{truepos+falseneg}$$(8)$$Precision=\frac{truepos}{truepos+falsepos}$$

Here, truepos is the number of intervals from a class that is classified correctly, falsepos is the number of intervals from another class that is classified incorrectly, and falseneg is the number of intervals belonging to a class that is classified as another class. A reminder is the proportion of time intervals belonging to a class that has been correctly classified, and accuracy is the proportion of intervals from the correct classification. We calculated the F-score of the classifier by averaging the individual F-scores four times. After obtaining the behavior datasets from the experiments, we used the k-fold cross-validation technique that splits them into *k (k* = *10)* sub-datasets. One of the *k* subsets is then utilized as the test set/validation set, while the remaining *k* − *1* subsets are combined to form the training set. The main dataset is divided into training and testing sub-datasets *k* times, with each data point appearing in the validation set exactly once and in the training set *k* − *1* times. At the end, the error estimation is averaged over all *k* trials to get total effectiveness of our model. Overall, we created the main dataset with 54,000 data points (9000 data points for each behavior) and split it into 10 sub-datasets (90% for the training, 10% for the testing dataset).

### 4.2. Classifiers Performances 

As aforementioned, we employed four different machine learning algorithms—SVM, KNN, DT, and GNB—to perform the behavior classification task. By comparing the accuracy level of these algorithms, we decided which algorithm is most suitable for the given job.

The second part of our experiment aimed to define the matter of the quaternion values to perform the behavior classification. Aiming for this, we created two different datasets with the same amount of data in each, where one of them consists of quaternion values of the dog’s movement and the other builds from the accelerometer and gyro sensor’s data. We ran the classifiers on each dataset and obtained their F-score performances. Finally, by choosing the highest F-score, we could conclude if quaternion values can lead to improvement of the classification.

The results of the behavior of the four classifiers evaluated using the F-score metric, recall, and accuracy are shown in Table 3. All four classifiers achieved acceptable results. At first glance, the overall results seem similar, but the F-scores of the activities are completely different. The SVM and GNB classifiers achieved the highest average accuracy level with 88% and 93%, respectively, whereas KNN and DT showed 86% and 82%. Among the available behaviors, the “resting” behavior was classified with 100% accuracy across all four classifiers, as during the resting period, the sensor’s data will not show any drastic changes that make it easy to classify the behavior. Moreover, the GNB classification results show the highest accuracy when classifying the standing and walking behaviors, with 94% and 86%, respectively. The remaining behaviors’ average accuracy levels are also the highest with the GNB classifier compared with SVM, KNN, and DT performances.

Furthermore, Table 3 shows the results of the classifiers’ performances with different datasets. The first dataset with quaternion values shows a higher average accuracy level than the second dataset, which consists of sensors’ data. As said above, the GNB classifier shows the highest level of average accuracy, and it is 93% when the first dataset is used as the primary data source for the model, whereas it is 88% with the second dataset. By observing the experiment results, we can ensure that computing the quaternion values of the movement serves to increase the classification accuracy.

In order to visualize and summarize the performance of the classification algorithms, we applied a confusion matrix to the experiment results. As can be seen from Figure 4, all classification algorithms recognized the “standing” and “resting” behaviors with high accuracy. Besides, the metrics indicated that, in the GNB classifier, the dataset with quaternion values has the highest level of accuracy compared with the dataset with sensor data.

### 4.3. Performance Comparisons

In order to detect how good the classifier’s performance is, we compared the accuracy of our classification with other studies that target to recognize the dog’s behaviors. However, several problems inhibited us from making a fair comparison:Types of dog breeds and their sizes are different across studies.The number of behaviors is different.The diversity of the metrics used to evaluate the model’s performance.

Therefore, in Table 4, we briefly compare the results of previous relevant work with our proposed method. Most references use a three-axis accelerometer and gyroscope data as the main data source to obtain the behavior classification, yet only a few have considered fusing the sensor’s data with video data. We see that the result of Chambers et al. [30] shows a quite similar accuracy level with our result; however, they confirmed accuracy in production via user validation, not by using any ML classifier outputs. Therefore, we observe that the performance of the proposed approach, which considers the calculation of quaternion values based on the sensors’ data, overcomes most of the previous studies’ results.

## 5. Conclusions

Four different classification representations for six types of dog behaviors (standing, walking, running, sitting, lying down, and resting) are presented in this study using wearable sensory technologies and ML algorithms. The proposed method, which uses a three-dimensional quaternion calculation with raw data, is well applicable to all four classification, training, and testing procedures. However, various studies on activity classification based on accelerometers and gyroscopes have used only raw data from sensors [34,37]. Usually, the raw data include data from three accelerometers and three gyroscopes. In our proposed method, the more meaningful raw data were extracted from the sensors. We applied the proposed method with four ML algorithms: KNN, RBM core SVM, DT, and GNB. The GNB classification provided the highest classification accuracy for dog behaviors based on the proposed method. For three classes of behavior (standing, running, and lying down), the classification achieved a better F-score than for the other two classes.

The accuracy of the trained model was proved during the experiments. The employment of the quaternion values led to an increase of 93% on average, whereas it was only 88% when only sensor data was considered the primary data source. The authors believe that the current approach can improve classification accuracy in real-time deployment scenarios and has a significant potential to improve dog welfare. The proposed method is mainly targeted for use in pet care systems to help maintain real-time activity status with high accuracy.

However, this paper has some limitations to the proposed approach. The proposed way of computing the quaternion values is based on combining the accelerometer and gyro sensors’ data, resulting in drifting that reduces the accurate measuring of rotations. We are trying to find a way to mitigate the drifts and hope to achieve a higher classification accuracy level. Second, this study considers only four different classification algorithms, and it may be interesting to see other classifiers’ performances. Since we experimented on dogs, most of them of a middle age and size, we would like to see our approach’s performance on small-sized dogs as the movement calibration can be sensitive to the body size. Finally, experiments on other animals with a different types of wearable devices could also lead to higher behavior accuracy. All of these things can be done in our future work.

## Figures and Tables

**Figure 1 sensors-22-09471-f001:**
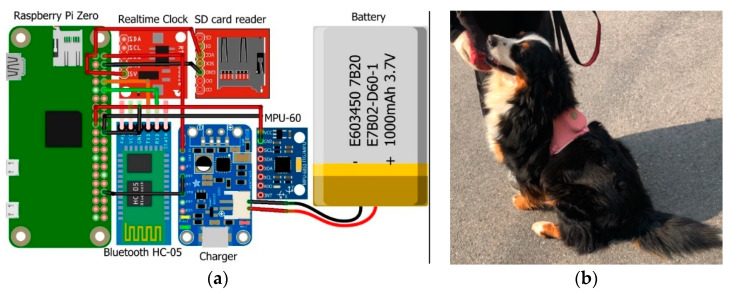
Smart costume: (**a**) The main parts of the smart costume are shown. They are Raspberry Pi Zero, RTC module, HC-05 Bluetooth module, MPU-6050 sensor, SD card reader, 3.7 v Li-Ion battery, and charger. (**b**) The fully constructed smart costume on a dog.

**Figure 2 sensors-22-09471-f002:**
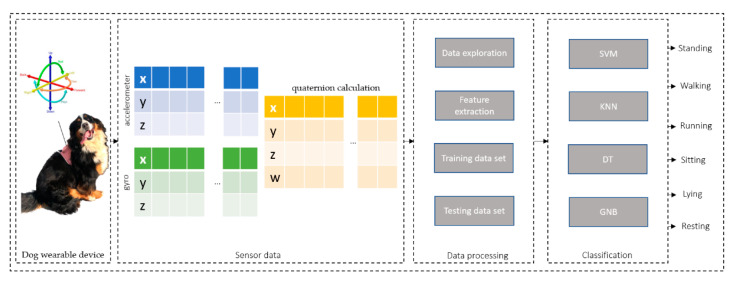
General process of the activity recognition system based on high-dimensional data.

**Figure 3 sensors-22-09471-f003:**
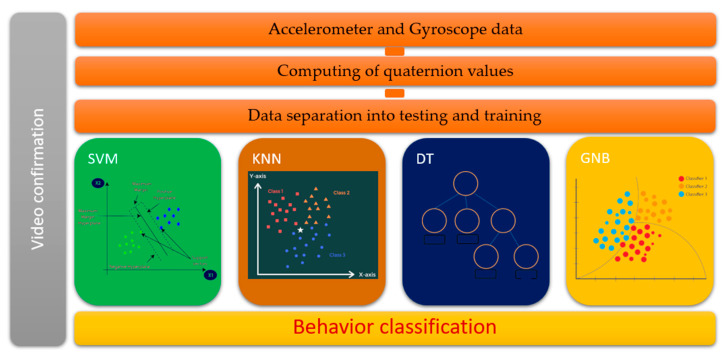
Experimental process model structure for all three classifications.

**Figure 4 sensors-22-09471-f004:**
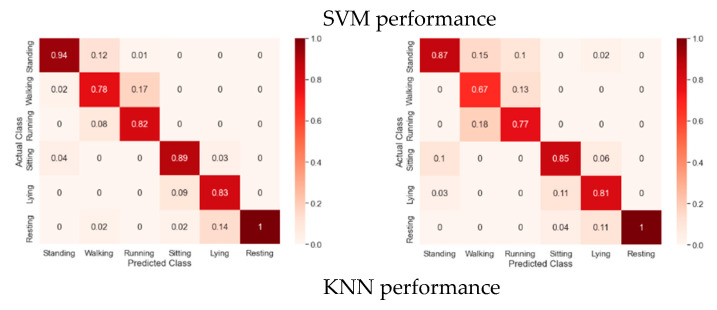
Confusion matrixes of classifiers performances. The left-side pictures represented the performance of a particular classifier when it used the quaternion values as the primary data source. The right-side pictures represented the performance of a particular classifier when it used the sensor data as the primary data source.

**Table 1 sensors-22-09471-t001:** Ethogram of the behaviors included in this study.

Class	Description
Standing	A normally standing dog has a four-legged standing posture in which it touches the ground and holds its head high.
Walking	A moving dog takes steps with its head upright. The dog’s legs are moved and lifted in the following order: steps are taken from the right front to the left rear, as well as from the left front and right rear legs.
Running	A fast-moving dog with frequent shaking.
Sitting	The dog has four extremities and its back on the ground, chest up. The angle between the chest and the ground is about 45°.
Lying	The dog has body extremities and chest touching the ground.
Resting	The dog has extremities of the body either on the right or the left side.

**Table 2 sensors-22-09471-t002:** List of house dogs that participated in this research.

Breed	Number	Age (Years)	Weight (kg)
Donggyeongi	3	3.2	13–14
Pungsan	2	3	13–18
Nureongi	2	4	14–17
Jindo	3	5	15–18
Bankar	2	4	12–15
Mixed breed	5	3–4	14–17
Bulgae	1	4	16

**Table 3 sensors-22-09471-t003:** Accuracy comparison of classifiers with two different datasets (dataset with quaternion values, dataset with sensors’ data).

SVM Performance						
	Results with Quaternion Data			Results with Sensors’ Data		
	Precision	Recall	**F-score**	Precision	Recall	**F-score**
Standing	0.95	0.93	**0.94**	0.91	0.84	**0.87**
Walking	0.73	0.85	**0.78**	0.61	0.75	**0.67**
Running	0.89	0.76	**0.82**	0.81	0.74	**0.77**
Sitting	0.91	0.89	**0.89**	0.86	0.84	**0.85**
Lying	0.89	0.79	**0.83**	0.83	0.77	**0.81**
Resting	1	1	**1**	1	1	**1**
**Average**	0.89	0.87	**0.88**	0.87	0.82	**0.83**
**KNN performance**						
	Results with quaternion data			Results with sensors’ data		
	Precision	Recall	**F-score**	Precision	Recall	**F-score**
Standing	0.79	1	**0.88**	0.71	0.89	**0.78**
Walking	0.76	0.73	**0.75**	0.69	0.70	**0.69**
Running	0.91	0.69	**0.79**	0.85	0.61	**0.71**
Sitting	0.89	0.88	**0.88**	0.82	0.80	**0.81**
Lying	0.88	0.80	**0.83**	0.83	0.76	**0.79**
Resting	1	1	**1**	1	1	**1**
**Average**	0.87	0.85	**0.86**	0.87	0.82	**0.79**
**DT performance**						
	Results with quaternion data			Results with sensors’ data		
	**Precision**	**Recall**	**F-score**	**Precision**	**Recall**	**F-score**
Standing	0.88	0.83	**0.85**	0.81	0.77	**0.79**
Walking	0.61	0.85	**0.71**	0.59	0.80	**0.68**
Running	0.88	0.58	**0.70**	0.82	0.55	**0.66**
Sitting	0.86	0.84	**0.85**	0.84	0.81	**0.82**
Lying	0.82	0.80	**0.81**	0.79	0.77	**0.78**
Resting	1	1	**1**	1	1	**1**
**Average**	0.84	0.81	**0.82**	0.87	0.82	**0.79**
**GNB performance**						
	Results with quaternion data			Results with sensors’ data		
	**Precision**	**Recall**	**F-score**	**Precision**	**Recall**	**F-score**
Standing	0.96	0.92	**0.94**	0.90	0.86	**0.88**
Walking	0.85	0.87	**0.86**	0.81	0.82	**0.81**
Running	0.95	0.93	**0.94**	0.89	0.85	**0.87**
Sitting	0.89	0.90	**0.89**	0.84	0.83	**0.83**
Lying	0.95	0.96	**0.95**	0.88	0.90	**0.89**
Resting	1	1	**1**	1	1	**1**
**Average**	0.93	0.93	** 0.93 **	0.87	0.82	**0.88**

**Table 4 sensors-22-09471-t004:** Performance comparison with other studies results.

**Studies**	**Year**	**Data Acquisition**	**Number of Behaviors**	**Accuracy**
Ladha et al. [25]	2013	3-axis accelerometer	17	69.6%
den Uijl et al. [26]	2017	3-axis accelerometer	8	92%
Gerencsér et al. [27]	2013	3-axis accelerometer, 3-axis gyroscope	7	>80%
Hoffman et al. [29]	2018	3-axis accelerometer	1	86%
Ferdinandy [28]	2020	3-axis accelerometer, 3-axis gyroscope	8	60–80%
Chambers et al. [30]	2021	5000 videos of more than 2500 dogs	7	30–95%
Kim et al. [31]	2022	Video, accelerometer, gyroscope	7	91%
**This paper**	2022	3-axis accelerometer, 3-axis gyroscope	6	86–95%

## Data Availability

Not applicable.

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
