# Peer review of "Enhanced Classification of Dog Activities with Quaternion-Based Fusion Approach on High-Dimensional Raw Data from Wearable Sensors"

_sensors, 2022, doi:10.3390/s22239471_

Round 1

Reviewer 1 Report

The paper presents an enhanced classification for dog activity recognition. The authors employ wearable sensors and apply quaternion-based fusion approach on high-dimensional raw data. The manuscript is well written and easy to follow. The following comments can improve the quality of the paper:

- Quaternion-based sensor fusion literature needs improvement. There are several works that use the aforementioned technique to improve the classification results and that literature review part is missing. 

- Authors use data originating from dogs with similar size and age. Age and size are factors that affect a dog's movement profile. The authors evaluate their technique on a larger variety of dogs.

- There is no comparison with existing techniques. Authors should evaluate their approach on public datasets and provide a comparison with benchmark/state-of-the-art papers.

- How and why the benchmarked classifiers were chosen?

- Authors should avoid using the word "Total accuracy" on results tables to present the total precision / recall /F1, as accuracy is a metric and may create confusion for the reader.

Author Response

Reviewer#1, Concern # 1: Quaternion-based sensor fusion literature needs improvement. There are several works that use the aforementioned technique to improve the classification results and that literature review part is missing.

Author response: Thank you for the valuable suggestion. We looked through the related works and created a new section dedicated to relevant literature reviews. Additionally, we reviewed several state-of-the-art papers with similar research directions and different approaches.

All changes are in Section 2 (page #3) and highlighted.

_____________________________________________________________________________________

Reviewer#1, Concern # 2: Authors use data originating from dogs with similar size and age. Age and size are factors that affect a dog's movement profile. The authors evaluate their technique on a larger variety of dogs.

Author response: We agree with the reviewer's opinion. Since the current study mostly relies on the quaternion values, representing the body rotations, the size of the dog's body is essential. Therefore, the current technique is most suitable for dogs of medium and large sizes. Regarding small dogs, we are currently working on experimenting with a new approach that can overcome the most recent studies.

This limitation has been highlighted in the Conclusion section, along with others.

_____________________________________________________________________________________

Reviewer#1, Concern # 3: There is no comparison with existing techniques. Authors should evaluate their approach on public datasets and provide a comparison with benchmark/state-of-the-art papers.

Author response: Thank you for your valuable comment. As mentioned in the manuscript, one of the objectives of this study is to detect whether the calculation of quaternion values and its consideration for behavior classification can lead to achieving accurate classification. Therefore, to show the proposed approach's preferability, we added Table 4, which displays general comparative information of previous studies, including when and by whom it was conducted, what method of data acquisition was used, number of targeted behaviors, and what level of accuracy was achieved.

We updated the manuscript by adding a new sub-section describing Table 4. All changes are highlighted.

_____________________________________________________________________________________

Reviewer#1, Concern # 4: How and why the benchmarked classifiers were chosen?

Author response: We appreciate your comment. As mentioned in the manuscript, we aim to recognize 6 number of dogs’ behaviors. Therefore, we used the classifier that can handle multi-class problems. Moreover, in order to detect the most suitable classifier algorithm, we employed the most common multi-class algorithms (SVM, KNN, decision tree, and GNB) and compared their performances by looking at their F-scores.

The same explanation is added in Section 3, Behavior classification sub-section, and is highlighted.

_____________________________________________________________________________________

Reviewer#1, Concern # 5: Authors should avoid using the word "Total accuracy" on results tables to present the total precision / recall /F1, as accuracy is a metric and may create confusion for the reader.

Author response: Thank you for pointing out the weakness part of the manuscript. We updated the table information by changing the word “Total accuracy” to “Average” as it indeed describes the average value of the accuracy.

Reviewer 2 Report

In this manuscript, the authors proposed a new feature extraction method based on quaternion values for dog activity/behavior recognition. Accelerometers and gyroscopes are used from wearable devices and classification models are trained using supervised machine-learning techniques. The dataset and low-cost device designed are strengths. The results seem reliable. However, there are several major concerns about manuscript structure that need to be resolved before publication:

[Major points]

1 - The description of the quaternion values calculation method should be better described in section 2.2. Reference 17 must be incorrect regarding the citation of the quaternion technique. My suggestion is to create a new section dedicated to the calculation of quaternation values.

2- Related works are few. Would you be able to show more recent state-of-the-art work? (2021, 2022 year). Suggestion: Create a related works section. It would be good to discuss which works have already successfully applied the quaternion technique and in which type of application?

3- I understood that the evaluation process consisted of an 80/20 hold-out(just one train-test split). Would you be able to show the classification results using cross-validation? Your dataset is not large and the cross-validation gives your model the opportunity to train on multiple training/testing splits. In addition, you can present a report variability of these model performances.

4- Would you be able to show the improved accuracy of your method by comparing your detection result with the state-of-the-art results?

[Minor points]

1- On pages 3 to 7, section 2.2 - after Table 1, paragraphs must be justified (this includes texts in bullets and captions);

2- References 17 and 26 are duplicated.

3- Data availability statement - Provide details on where data supporting the reported results can be found, including links to publicly archived datasets analyzed or generated during the study. Reproducibility is important.

Author Response

Reviewer#2, Concern # 1: The description of the quaternion values calculation method should be better described in section 2.2. Reference 17 must be incorrect regarding the citation of the quaternion technique. My suggestion is to create a new section dedicated to the calculation of quaternation values.

Author response: Thank you for the valuable suggestion. We corrected the citation and updated the manuscript according to the reviewer’s recommendation.

All changes are in Section 3 (page #5) and highlighted.

_____________________________________________________________________________________

 Reviewer#2, Concern # 2: Related works are few. Would you be able to show more recent state-of-the-art work? (2021, 2022 year). Suggestion: Create a related works section. It would be good to discuss which works have already successfully applied the quaternion technique and in which type of application?

Author response: Thank you for the valuable suggestion. As reviewer’s request, we looked through the related works and created a new section dedicated to relevant literature reviews. Additionally, we reviewed several state-of-the-art papers with similar research directions and different approaches.

All changes are in Section 2 (page #3) and highlighted.

_____________________________________________________________________________________

Reviewer#2, Concern # 3: I understood that the evaluation process consisted of an 80/20 hold-out(just one train-test split). Would you be able to show the classification results using cross-validation? Your dataset is not large, and the cross-validation gives your model the opportunity to train on multiple training/testing splits. In addition, you can present a report variability of these model performances.

Author response: We appreciate your comment. We re-train the classifier models by changing the previous technique (split the dataset with an 80:20 ratio for training and testing sub-datasets) with the requested k-fold cross-validation technique that divides the main dataset into k sub-datasets (in our case, k=10). One of the k subsets is then utilized as the test set/validation set, while the remaining k-1 subsets are combined to form the training set. The main dataset is divided into training and testing sub-datasets k times, with each data point appearing in the validation set exactly once and in the training set k-1 times. At the end the error estimation is averaged over all k trials to get total effectiveness of our model.  

All changes are in Section 4 (page #8) and highlighted.

_____________________________________________________________________________________

Reviewer#2, Concern # 4: Would you be able to show the improved accuracy of your method by comparing your detection result with the state-of-the-art results?

Author response: Thank you for your valuable comment. As mentioned in the manuscript, one of the objectives of this study is to detect whether the calculation of quaternion values and its consideration for behavior classification can lead to achieving accurate classification. Therefore, to show the proposed approach's preferability, we added Table 4, which displays general comparative information of previous studies, including when and by whom it was conducted, what method of data acquisition was used, number of targeted behaviors, and what level of accuracy was achieved.

We updated the manuscript by adding a new sub-section describing Table 4. All changes are highlighted.

_____________________________________________________________________________________

Reviewer#2, Concern # 5: On pages 3 to 7, section 2.2 - after Table 1, paragraphs must be justified (this includes texts in bullets and captions)

Author response: Thank you for the correction. We looked through the manuscript and corrected the above-mentioned minor points across the document.

_____________________________________________________________________________________

Reviewer#2, Concern # 6: References 17 and 26 are duplicated.

Author response: Thank you for pointing out mistake. We corrected the mistake and reviewed other related works as well.

All changes are in the References section (page #13) and highlighted.

_____________________________________________________________________________________

 Reviewer#2, Concern # 7: Data availability statement - Provide details on where data supporting the reported results can be found, including links to publicly archived datasets analyzed or generated during the study. Reproducibility is important.

Author response: We appreciate your comment. This study is considered one part of the big project called “Smart Pet”. This project involves other important parts, and we are working on them individually. Since the project isn’t finished, we are not allowed to share the dataset or the other parameters regarding the project yet. As soon as we finish our task, we plan to include the relevant data information in the final paper.

Round 2

Reviewer 1 Report

The authors addressed all the comments. Therefore, the manuscript is acceptable in its present form.

Reviewer 2 Report

The authors properly replied to my concerns. I have no other questions left.